# Resident Willingness to Pay for Ecosystem Services in Hillside Forests

**DOI:** 10.3390/ijerph19106193

**Published:** 2022-05-19

**Authors:** Wan-Jiun Chen, Jihn-Fa Jan, Chih-Hsin Chung, Shyue-Cherng Liaw

**Affiliations:** 1Department of Economics, Chinese Culture University, No. 55, HawKang Rd., Taipei 111, Taiwan; cwj@ulive.pccu.edu.tw or; 2Department of Land Economics, National Chengchi University, No. 64, Section 2, Zhinan Rd., Taipei 116, Taiwan; jfjan@nccu.edu.tw; 3Department of Forestry and Natural Resources, National Ilan University, No. 1, Section 1, Shennong Rd., Yilan City 260, Yilan County, Taiwan; chchung@ems.niu.edu.tw; 4Department of Geography, National Taiwan Normal University, No. 162, Section 1, Heping E. Rd., Taipei 106, Taiwan

**Keywords:** forests, ecosystem services, local industries, economic value, organic farming, contingent valuation method

## Abstract

This study investigated the willingness of residents to pay for ecosystem services in a hillside forest in the Lanyang River Basin, which is among the most vulnerable watersheds in Taiwan. The economic value of provisioning, regulating, cultural, and supporting ecosystem services was evaluated. The Contingent Valuation Method (CVM) was applied for economic analysis of public welfare. The determinants of the economic values were identified. A total of 444 respondents completed the questionnaire. The results revealed that the four ecosystem services had high economic value, indicating that conserving hillside forests can ensure the welfare of nearby residents. The findings of this study can serve as reference for regional land planning and social and economic system development policies. In addition, this study addressed policy implementation from the perspective of ecological economics to contribute to an improved Anthropocene.

## 1. Introduction

Our planet is an ecosystem. The successive evolution of our world has transformed from an empty world to a full world, as defined by ecological economists [1,2,3,4,5]. It is possible to conceive that our world is highly developed; however, the planetary ecosystem and resources are finite. Because the economic system is a subsystem in the planetary boundary [6,7], the finite nature of Earth’s resources limits the economic growth.

Anthropocene is often used to represent the status quo of our current full world [8,9,10,11]. In previous studies, Bennett et al. [9] suggested the needs of initiatives that can lead to a better Anthropocene, and McPhearson et al. [11] also point out radical improvements are required for a better Anthropocene.

In philosophy, anthropocentrism refers to the belief that humans are the central or the most significant entity in the world. In accordance with the anthropocentric viewpoint, diverse functions performed under the circulation of the ecosystem directly or indirectly provide local residents with tangible or intangible benefits. The benefits obtained by individuals from the ecosystem refer to ecosystem services. The Millennium Ecosystem Assessment (MEA) was conducted by the United Nations to examine the effects of changes in the ecosystem on the well-being of humans. In the MEA, the relationship between ecosystem services and human well-being was determined by dividing ecosystem services into four aspects: provisioning, regulating, cultural, and supporting [12]. Provisioning services refer to material products that individuals obtain from the ecosystem, including food, fresh water, and biofuels. Regulating services refer to benefits obtained from physical, chemical, and biological processes in nature, including natural disaster control, carbon storage, climate and water regulation, and pest and disease control. Cultural services are defined as intangible and nonmaterial benefits provided by the ecosystem to humans, including spiritual, aesthetic, educational, and recreational values and cultural diversity. Supporting services refer to indirect services provided by the biochemical cycles of ecosystems; these services are necessary for the production of provisioning, regulating, and cultural services, including fundamental processes that support soil formation, nutrient cycling, photosynthesis, biodiversity, hydrology, and nutrient cycling on which humans are highly dependent [12,13]. The multifaceted and critical functions of the ecosystem are highly valuable and closely related to the well-being of individuals living in that ecosystem [14]. Only parts of these services are traded in the market system.

Direct benefits obtained by individuals from commercialized goods that are produced from the ecosystem and traded services that are provided by the ecosystem can be clearly determined. Moreover, the price in the trading systems usually serve as determinants and the driving forces that guide people’s decisions in our society. However, the price in the trading market system can only partially reflect the value of ecosystem services to humans. That the values in the market trading system cannot represent the full value of ecosystems to humans is proposed in the perspective of ecological economics [7].

In terms of environmental protection and conservation, if the value of each service provided by the ecosystem cannot be clearly identified and quantified, the intangible value that has not been quantified may be misunderstood and ignored in mainstream society, resulting in the loss of crucial ecosystem services as a result of lacking awareness. Therefore, scholars have clearly defined specific ecosystem services [12] and evaluated the value of ecosystem services [7].

Because of the complexity of ecosystem services, existing market prices cannot precisely quantify their value. In the long-term pursuit of economic growth, mainstream capitalized society has lost the balance between the natural society and human society, neglecting the coexistence of the ecological environment. From the transdisciplinary perspective of ecological economics, after economic development for a certain period, the world has transitioned from an empty world to a world full of humans and economic activities [1,2]. The space for human life and economic activities has expanded from relatively safe and flat land to hillside areas with extremely high environmental risks. Further reclamation or development can increase the multifaceted risks of hillsides, resulting in their increased sensitivity to extreme climate events such as heavy rainfall and floods. The prioritization of income growth by mainstream society has reduced the necessity and weakened the coercive and driving forces of environmental conservation and climate adaptation by the government and relevant private sectors. Assessment of the economic value of marketable and nonmarketable ecosystem services can help identify the value indicators of the four ecosystem services. Moreover, the findings of such assessment can guide the transformation of policies and strategies in the mainstream economy. On the basis of anthropocentric philosophy, the economic valuation techniques developed in the field of neoclassical economics. The assessment helps in guiding whether to conserve or exploit nature for human well-being is based on the benefit to human beings [15].

The boundary between human society and nature is the critical frontier requiring government intervention for mediation. Well-maintained forests in hillside areas enable local residents to reap benefits from ecosystem services. Hillslope lands are vulnerable to climate change and human activities [16,17]. Human disturbance can aggravate landslides and water and soil pollutions [18]. These problems are especially acute in mountainous areas in Taiwan.

Concerns regarding the conservation and protection of ecosystem services exist worldwide, and land-use changes might hinder optimal use of local ecosystem services [19]. Land-use zoning generally serves as the basis for ecological plans [20]. The Taiwanese government has planned land zoning on the basis of regional landscape characteristics in Taiwan [21]. By using the Lanyang River Basin in Yilan County as the study site, this study evaluated the economic value of ecosystem services available in conserved low-altitude hillside forest areas, which are classified by government land classifications.

Residents were interviewed, using a questionnaire survey, to determine their opinions regarding the four ecosystem services (provisioning, regulating, cultural, and supporting) provided by the adjacent forest on the low-altitude hillside of the Lanyang River Basin. The benefits of the ecosystem services were determined using a five-point Likert scale, and the economic value of the ecosystem services was determined using the single-bounded dichotomous contingent valuation method.

The findings of this study can provide policy references for guiding the formulation and promotion of local economic development strategies, enhancing local resilience to future impacts, and reducing social and economic damage caused by inappropriate land use.

## 2. Materials and Methods

### 2.1. Study Site

The climate in Taiwan ranges from tropical to subtropical. Being surrounded by sea, Taiwan frequently experiences seasonal monsoons and severe typhoons in summer and autumn. Climate change has resulted in extreme rainfall, drought, and strong winds. In Taiwan, the mountains are high and steep, and the coastal plains, occupying a small area, are densely populated. The marginal land area is rich in natural resources. Prudent land classification and corresponding zoning to preserve forests and prevent disasters have resulted in the regulation of land use and protection of vulnerable hillside areas. Furthermore, the intervention of the government to limit the development of sloping land resources and protect steep slopes from large-scale disasters has improved the conservation of soil and water, the integrity of the overall ecosystem of the sloping land, and the protection of existing ecosystem services in hillside areas.

The impact of climate change has caused an increase in the frequency of extreme climate events and changes in the distribution patterns of high temperature and rainfall. The United Nations Intergovernmental Panel on Climate Change (IPCC) synthesized the latest research in various academic fields in the 2019 Special Report on Climate Change and Land [22], discussing climate change, land degradation, sustainable land management, food security, and greenhouse gas fluxes in terrestrial ecosystems. In the report, the IPCC indicated that more than a quarter of the world’s land is facing the risk of degradation and that climate change continues to intensify; thus, strengthening the protection and restoration of forests is a key solution to prevent land degradation and disasters [22].

Extreme weather events resulting from climate change often cause substantial damage in many areas of Taiwan, especially land and rock disasters on slopes caused by high-intensity typhoon rainfall. Global warming has caused extreme climate uncertainty, changes in precipitation patterns, and drought and flood problems. Moreover, unusual weather caused by climate change is frequently reported in Taiwan and worldwide, breaking meteorological records and posing a considerable threat to a country’s social economy, individuals, and property. Thus, responding to catastrophic weather has become a crucial problem in terms of national security. Uncertainty regarding climate such as uneven rainfall in the monsoon and typhoon seasons poses a particular challenge to Taiwan’s slope land areas [23,24,25,26].

The gradual movement of some agricultural activities from plains to mountainous areas would exacerbate the fragmentation of the hillside landscape. Moreover, this situation would result in the hillside and its ecosystem being divided into scattered units, thus reducing the services of the ecosystem and increasing the complexity of hillside disasters. Therefore, determining the value of the ecosystem services that can be destroyed by natural disasters caused by climate change is essential, especially in the context of misleading policies or poor preparation for mitigating future climate change. However, low-altitude hillside areas, which are located at the critical interface between man and nature, are directly affected by human activities. Because of climate change, fruits and vegetables are increasingly being cultivated in more temperate, cooler, and well-drained mountainous areas [27,28]. The optimal temperature for the growth of cabbage, which is widely cultivated in Taiwan, is approximately 20 °C. However, cabbage grows slowly and its taste is negatively affected during high temperatures in summer that exceed 30 °C. Unfavorable conditions resulting from climate change has forced the movement of agriculture to mountainous areas [27,28]. 

Against the backdrop of potential threats and crises, the sustainable management of the ecosystem in mountainous areas has become crucial. Determining the economic value of the ecosystem services provided by forests to residents can inform the development of policies to manage possible changes in the relationship between humans and nature in the future.

The Lanyang River Basin is located in Yilan County, northeastern Taiwan, and covers plains and high mountainous areas. This basin is rich in biodiversity and is a crucial area for agricultural production and recreational activities. The Lanyang River Basin frequently experiences strong winds, torrential rains, and typhoons, which adversely affect the ecosystem and environment. Climate change, extreme weather events, typhoons, and torrential rain may lead to the loss of soil and water resources and the destruction of sloping animal and plant habitats, posing a potential threat to the ecosystem services. In Taiwan, Yilan County is among the areas most affected by typhoons and is thus a suitable representative research area for investigating the impact of climate change on hillsides.

By using the Lanyang River Basin as the research area (Figure 1), this study analyzed the current economic value of the ecosystem services in low-altitude hillside areas to provide basic background knowledge for the management and adjustment of catchment areas.

Three of Yilan County’s 12 townships are located at a lower altitude, a conjunct area between the Lanyang Plain and mountains, and residents live in the vicinity of the hillside. Two townships are located high on the watershed. The remaining townships are located close to the sea and have active economic activities.

In this study, the three townships located at low altitude were investigated: Sanshing Township (SST), Yuangshan Township (YST), and Dongshan Township (DST). These three townships are geographically located at the intersection of mountains and plains, with areas of 74.77, 133.69, and 78.02 km^2^, respectively, and population densities of 21,221, 32,177, and 52,954, respectively. In the following subsections, the basic characteristics of these three townships are described on the basis of information provided by Yilan County Government, Taiwan [29].

#### 2.1.1. SST

SST is located to the west of the Lanyang Plain. This township contains many plains and river networks; the rocks and sand coming down from the fragile mountainous area are accumulated in riverbeds. The land SST is located on was formed following the creation of a large sandbar in the creek bed of the Lanyang River. In the past, SST was subject to frequent river flooding, and new residents arrived to reclaim the land. This township is characterized by frequent flood damage, and the principal crops were rice, peanuts, and sugarcane. 

SST is prone to flooding due to its geographical location. Moreover, individuals of multiple ethnic groups reside in SST. Trade mainly occurs between individuals living in mountainous areas and individuals living on the plains. Thus, SST has been a crucial place for multiethnic and cultural fusion since ancient times and a place of ethnic and cultural conflicts and trade. 

In recent decades, floods have become less frequent due to intervention by the government, and ethnic conflicts between aboriginal and Han people have been minimized following ethnic diffusion; however, the stagnation of social and economic development has prompted the exodus of many residents. Villages are now home mostly to older people and children. However, due to dedicated agricultural development in SST in the last two decades, it has attracted attention for its high-quality agricultural produce. The cultivation of green onion, garlic, silver willow, and pear has provided economic benefits to the residents and improved their lives.

#### 2.1.2. YST

YST is mostly a hilly area with a beautiful landscape. The sloping topography and climatic conditions of YST are suitable for the cultivation of various crops such as tangerines, pineapples, bamboo shoots, ginger, starfruit, leeks, guava, lotus mist, shallots, and pears. The green hills and fruit fields are particularly suitable for the promotion of agricultural recreational activities. Fruit picking services are provided during the ripening season.

#### 2.1.3. DST

DST is mostly a hilly area, and tea plants and fruits are mainly cultivated in this area. In DST, agriculture is actively transforming into precision farming to promote local agricultural recreational activities, including tea gardens, local food restaurants, and sightseeing orchards. Tea is the most popular agricultural produce. With the efforts of local farmers’ associations, the cultivation of local specialty products, such as tea, pears, pomelo, peach, and yam, is promoted. Agricultural recreational activities involve recreational forest trails in mountainous areas and visits to waterfalls, lakes, flood diversion weirs, and tree seedling nurseries.

### 2.2. Methodology

In Taiwan, because of high mountains with steep slopes, short rivers, rapid instream flow, and heavy rainstorms, dangerous hillsides and geologically fragile zones are challenging to exploit. However, long-term soil conservation practices have effectively protected hillside forest areas.

The hillside forest ecosystem provides ecosystem services for local residents, and the contingent valuation method based on consumption theory can be used to estimate the economic value of marketable and nonmarketable goods [30]. The economic value can be used as an indicator to reflect the economic value of the ecosystem services [31,32].

This study evaluated the economic value of the ecosystem services of the hillside areas of the Lanyang River Basin forest to local residents.

Ecosystem services provide both marketable and nonmarketable benefits, and their value cannot be fully and directly presented in the trading market. To determine the complete economic value of the four specific ecosystem services for residents, with the aim of facilitating government decision-making, this study used the single-bounded dichotomous contingent valuation method. This study established a hypothetical market in which respondents needed to indicate their willingness to pay a certain amount for a given service. The economic value of the ecosystem services was analyzed using the single-boundary dual-condition evaluation method. This quantitative analysis can help identify factors that affect the willingness to pay for services and the relevant amount; this information can be beneficial for formulating relevant policies.

#### 2.2.1. Contingent Valuation Method

The contingent valuation method is an application of stated preference that involves (1) establishing a hypothetical market for target goods or services to be evaluated, (2) eliciting values by directly asking stakeholders, and (3) calculating the economic value of the target goods or services based on the theory of welfare economics. This method is used to evaluate various goods and services and is often used to examine nonmarketable goods and services. In the hypothetical market, various elicitation methods are used in the questionnaire. To simulate the real purchasing pattern in the exchange market, closed-ended questions are preferred over open-ended questions. Cameron and James [33] indicated that the single-bounded method with close-ended questions is efficient. The binary choice method is regarded as an ideal method. The estimation process for dichotomous choices is widely applied for evaluating environmental goods and services [34].

In binary selection, if a series of higher or lower bidding amounts remain after the first binary selection, the single-bounded dichotomous method is extended to make it double-bounded, triple-bounded, and so on. The aforementioned methods have been compared in various empirical studies [35,36,37,38]. Boyle et al. [39] used the single-bounded method, whereas Kumaraswamy [40], Sundar and Subbiah [41], Kanninen [42], Yoo and Yang [43], and Yoo and Kwak [44] have used the double-bounded method. Langford, Bateman, and Langford [45], Bateman et al. [46], and Carson and Hanemann [38] have used triple-bounded dichotomous questions. Moreover, studies have discussed the limitations and solutions of the use of these methods [47,48,49] in terms of bias during the survey and the respondent’s psychological behaviors.

Although the increase in the bidding time in a questionnaire would shorten the interval used to estimate economic values, Hanemann et al. [50] theoretically verified that the estimation results of the double-bounded method were more statistically efficient than those of the single-bounded method. However, an increase in the bidding time would increase the complexity and difficulty in processes at both the stages of practically interviewing respondents and technically estimating the average willingness to pay. Increasing the number of the bidding inquiries involving binary choices would reduce the marginal efficiency of estimation [51].

The single-bounded dichotomous evaluation method has been employed to examine the economic value of ecosystem services in empirical studies [52,53,54,55] due to the simplicity and efficiency of the interview and analysis processes, despite considerable discussion of potential problems since its early use by Bishop, Heberlein, and Kealy [56]. Moreover, the evaluation has significance in the empirical application in ecology assessment [57,58,59,60]. Gould et al. [55] had indicated that the ecosystem services research and evaluations can advance ecological economics principles. Hence, the contingent valuation method remains useful for identifying environmental and ecological value indicators.

#### 2.2.2. Residents and Respondents

In this study, local residents were regarded as critical stakeholders because they directly received the ecosystem services from the hillside forest. The residents’ views on the economic value of the ecosystem services in the constructed hypothetical market were explored in this study. The residents’ willingness to pay for conserving the current ecosystem services generated from the forest present in the vicinity was evaluated.

#### 2.2.3. Eliciting Process

A closed-ended questionnaire with a single-bounded dichotomous choice method was used to elicit the willingness of respondents to pay for the service. This method is similar to the ordinary trading behavior of individuals in the market, and the answering of questions is easier for respondents in this method. Respondents were required to simply indicate whether they were willing to pay the assigned amount provided in the questionnaire. The following question was asked to the respondent: “With the knowledge that the present ecosystem services will be destroyed one day, would you be willing to pay $ A_i_^0^ to conserve your current ecosystem services now?” The dichotomous choices to be selected were “willing” or “unwilling” to pay. Such questions involving a binary response correspond to the single-bounded discontinuous contingent valuation method. If the *i*th response is “yes,” the *i-*th respondent’s real willingness to pay (A_i_) would be in the range (A_i_^0^, ∞); if the response is “no,” A_i_ would be in the range of (0, A_i_^0^).

#### 2.2.4. Embedding Effect

The embedding effect occurs when parts of inseparable mosaics are estimated. This effect is observed when commodities are analyzed on the basis of their characteristics [61].

When the economic value of an individual service is analyzed, the embedded effect is likely to occur because the four ecosystem services are simultaneously generated from the ecosystem and are not independent commodities. The ecological circulating and functioning systems for the four ecosystem services classified by MEA [12] are not mutually exclusive. The provisioning, regulating, culture, and supporting ecosystem services cannot be separate.

Although these services can be estimated separately, the overall ecosystem value cannot be represented by the linear addition of estimated individual values because of the presence of unknown intangible and complex interconnections in the system and between them. The four ecosystem services are embedded in each other. Thus, this study analyzed the economic benefits of the four ecosystem services individually.

#### 2.2.5. Payment Channel

A payment was asked for the ecosystem services to conserve the well-protected hillside forest. A well-designed payment method can promote and affect the willingness to pay and the estimated economic value of a service. In terms of payment methods, residents could use conservation funds from the local community. In addition, government taxations are usually regarded as a lump sum amount already paid for all general public services, and the inclusion of this payment channel in questionnaires tends to evoke protesting and invalid responses [62].

#### 2.2.6. Estimation Procedure of the Single-Bounded Method

This study used the single-bounded method reported by Hanemann [30] to estimate the economic value. The economic value was estimated by fitting questionnaire data to the binary logit function. The logit function is as follows:(1)$$P\left(Y\right)={\left(1+\exp^{-\left(\beta_{0}+\beta_{1}A+X\Phi \right)+e}\right)}^{-1}$$
where *P*(*Y*) is the probability of the respondent saying “yes;” *A* is the bidding value; *X* is a vector of independent variables including the demographic characteristics of respondents; $\beta_{0}$, $\beta_{1}$, and Φ are parameters; and *e* is the random error. Estimation of the logistic model requires random errors, implying the presence of incomplete knowledge on respondents’ preferences. 

The point estimate of the economic value was calculated with reference to the studies of Cameron [34,63] and after performing logistic regression (1) and determining the estimated coefficient of the bidding variable $\hat{\beta_{1}}$
(2)$$E(\mathrm{WTP})=-1/\hat{\beta_{1}}$$

#### 2.2.7. Pilot, Bidding Values, and Survey

To collect information on residents’ opinions and willingness to pay for the four ecosystem services, an on-site questionnaire survey was performed during August 2021 in SST, YST, and DST. The survey interviews were performed at the township office, post office, train station, and farmers’ association of these townships. A total of 450 residents were randomly selected, of whom 90, 136, and 224 were from SST, YST, and DST, respectively; the number of the township residents depended on the total population of each township. 

A total of 444 respondents completed the questionnaire, of whom 89, 136, and 219 were residents of SST, YST, and DST, respectively.

The overall bidding range and the seven bidding values were determined on the basis of WTP from the pilot by assuming that the population distribution of WTP could be reliably derived from the pilot survey. The five bidding values were 250, 500, 1000, 2000, and 3000 New Taiwan dollar (NTD), and they were randomly assigned to the respondents in the survey. 

The bidding value was set at 250, 500, 1000, 2000, and 3000 NTD in accordance with the findings of the pilot survey conducted from July to August 2021. In the pilot survey, 50 residents were asked an open-ended question to determine their willingness to pay for conserving current ecosystem services in the hillside forest. The bidding values in the final survey represent the 20th, 30th, 50th, 70th, and 80th percentile (P20, P30, P50, P70, and P80, respectively) of the amount indicated in responses to the open-ended question in the pilot survey (Table 1).

On the basis of the theoretical principle of welfare economics, utility occurs when an individual consumes a commodity or enjoys a service and is willing to exchange the benefit for a monetary amount. The welfare of the consumer can be calculated by following the principle of consumer theory. This study established a hypothetical market, and the residents were asked if they are willing to pay a bidding amount each year.

The bidding amounts of 250, 500, 1000, 2000, and 3000 NTD were randomly assigned to the questionnaire items. That is, the respondents simply responded to the value provided in the questionnaire item; they did not choose a specific bidding amount for each questionnaire item. The results of the survey, the statistics of responses to different bidding amounts, and the estimated economic value are reported in the next section.

## 3. Results

### 3.1. Residents’ Agreement with the Ecosystem Services

In the survey, the opinions of the residents regarding the ecosystem services provided by the nearby hillside forest were investigated. The interviewed residents exhibited high agreement with statements regarding the hillside forest ecosystem’s ability to provide the provisioning, regulating, cultural, and supporting ecosystem services (Table 2 and Table 3).

### 3.2. Response Statistics of Different Bidding Amounts

The different bidding amounts were distributed evenly (Table 4).

The interviewees were asked to indicate if they were willing to pay the assigned bidding amount. Respondents answered “yes” if their valuation of the ecosystem service was higher than the bidding amount and “no” if their valuation was lower than the amount asked. Table 5 presents the statistical data of the respondents’ responses; the number of positive responses slightly declined with the increase in the bidding amount in the three townships.

The results indicated that 60.67%, 58.82%, and 55.71% of the residents of SST, YST, and DST were willing to pay. Most of the respondents strongly agreed with the value of the ecosystem services provided by the hillside forest in the Lanyang River Basin. Nearly half of the respondents expressed that they were willing to pay the bidding amount.

### 3.3. Determinants of Residents’ Willingness to Pay

This study performed logit regression to analyze the determinants of the residents’ willingness to pay for the randomly assigned bidding amount. On the basis of the estimates of logit regression, the economic value of the ecosystem services was calculated by following Cameron [34]. The dependent variable *P*(*Y*) was the binary responses of the residents in the single-bounded contingent valuation method and calculated as follows:(3)$$P\left(Y_{i}\right)=f\left({\mathrm{BID}}_{i},AGREE_{EE_{i}},\;GENDER_{i},\;OCCU_{D_{PUBLIC_{i}}},\;STAY_{HR_{i}},\;AGE_{D60_{i}}*INC_{D30_{i}},\;INC_{10T_{i}},\;DON_{D_{i}}\right)+\mu_{i}$$
where μ is the residual and subscript *i* represents the *i*th respondent. Table 6 presents the definitions of the variables included in this regression and their descriptive statistics. In addition to the bidding variables, demographic variables have high accessibility and are often used as regressors. Other variables, namely the individual agreement with the value of the ecosystem services offered by the forest, length of stay per visit to represent the personal interaction with the hillside forest, and history of monetary donation for environmental protection, were included in logistic regression. The degree of the agreement of the respondents with statements regarding the value of the four ecosystem services was measured using a five-point Likert scale. Because the five points used in the Likert scale were not equidistant, its arithmetic mean is not indicated in the statistics in Table 6. Monetary donation for environmental protection, which is an actual economic response action, was included as a regressor.

Table 7 lists the estimated results of logistic regression. The extrapolation and interpolation results indicated significant variables in logistic regression. The variables listed in Table 7 were examined using the chi-square test and Student’s *t* test, and high collinearity was not observed between the explanatory variables. The chi-square test was used to examine associations between categorical variables, and Student’s *t* test was used to determine correlations between continuous variables. The determinants were determined on the basis of the estimates in Table 7.

In the estimated logit regression model, the bidding amounts assigned to the respondents (variable BID) were significantly negatively correlated with the four ecosystem services. The willingness to pay tended to decrease with the increase in the bidding amount. However, the bidding amount was not the only significant variable.

The willingness to pay increased with agreement regarding the value offered by the hillside forest in terms of the four ecosystem services.

The willingness to pay for conserving the regulating, culture, and supporting ecosystem services exhibited a significant positive association with length of stay per visit. Length of stay per visit was included in this study as a variable to represent personal direct interaction with the hillside forest because the regulating, culture, and supporting ecosystem services necessitate on-site interaction between humans and nature.

An individual’s judgment regarding the affordability of a service is constrained by income. Household income positively affected the willingness to pay the bidding amount.

The residents who were employed in the military, civil service, or education sector tended to reject the payment. Moreover, those who did not directly depend on local resources for their livelihood declined to pay for conserving the ecosystem services. Those who had high reliance on forest resources for their livelihood assigned higher value to the ecosystem services.

An individual’s willingness to pay for conserving the ecosystem services offered by the forest was strongly influenced by a history of donations for environmental protection. The residents who had ever donated for environmental protection were more willing to pay.

The older residents with low income tended to decline paying the bidding amount because of their perception of affordability and local economic stagnation. Rampant urbanization in Taiwan has prompted the exodus of people from rural areas despite local social and economic development initiatives promoted by the government [64].

The women were more willing than men to pay for conserving the regulating ecosystem services; willingness to pay for the other services did not differ significantly between the sexes.

### 3.4. Economic Value of the Ecosystem Services

This study examined the economic value of the ecosystem services on the basis of the results of the single-bounded dichotomous contingent valuation method and regression analysis (Table 7).

The economic value of the forest ecosystem services to residents was determined by evaluating the residents’ willingness to pay for conserving the current ecosystem services that they received and enjoyed. Some respondents psychologically resisted paying for the ecosystem services they already had access to; these should be classified as protesting samples. Those who did not agree with the economic value received no benefits from the ecosystem services. The protesting residents were unwilling to pay the bidding amount, indicating that the economic value they assigned to the service was lower than the bidding amount.

The point estimates for the economic value of the provisioning, regulating, culture, and supporting ecosystem services were 2688.17, 2092.05, 2577.32, and 2551.02 NTD, respectively (Table 8).

## 4. Discussion

### 4.1. Land Zoning Policy and Government Intervention

Taiwan is located at the junction of the Eurasian plate and the Philippine plate and has high mountains and steep slopes. The Lanyang River Basin, located in northeast Taiwan, is the most vulnerable watershed on the island. The stream, only 73 km long, originates from more than 3000 m above sea level from the northern Nanhu Mountain, to rush into the sea. It flows through the entire Yilan County. The watershed covers an area of 978 km^2^ [65]. In the middle- and high-altitude areas of the Lanyang River Basin (areas with an altitude of approximately 500 m above sea level), the mountains are high and steep, with limited human activities. Where the Lanyang River flows into the foothills in its downstream reaches, local residents are typically involved in agricultural activities. The river then enters the delta plain, which is metropolitan area with numerous human activities.

The land zoning plan implemented by the government calls for forest coverage in the foothills of the Lanyang River Basin, and local ecosystem services are relatively effective. However, the basin is affected by land-use patterns such as agricultural reclamation. 

Because the entire water catchment area is located in the northeast of Taiwan and it is on the windward side of most typhoons striking the island, the land is highly vulnerable. Thus, this land has been the focus area of global environmental changes. The benefits and impacts of low-altitude piedmont forest ecosystems on local residents merit in-depth study and evaluation and are directly related to the current agricultural reclamation development and climate change adaptation. Further human exploitation would result in an increase in the risk of disasters.

The evidences provided in this study affirmed the need to conserve hillside forests as the residents have high agreements toward the four aspects of the ecosystem services (Table 2 and Table 3).

Moreover, with the increasing threat of natural disasters due to climate change, information on the value of the ecosystem services can be used as a policy basis for the management of low-altitude sloping land disaster risk systems. The economic values of the ecosystem services the residents received are evidenced high as indicated in Table 8. This information can promote the establishment of disaster adaptation strategies for maintaining the hillside ecosystem and its services as well as facilitate the overall spatial planning for hillside areas under climate change.

### 4.2. Better Governance of Our Full World at Current Anthropocene

Based on the image of our current world—a full world, as it is called by Herman Dally [1]—it tends to be a world of anthropocentric Anthropocene, and humans are the primary benefactors and the receivers of ecosystem services [66,67]. Hence, the present study assessing for a relevant value indicator that can serve as an index for government interventions would help a better governance of our full world at current Anthropocene. The economic values of provisioning, regulating, cultural, and supporting ecosystem services enhance the feasibility of policy formation for land zoning at mountain frontiers in an ultimate use of the ecosystem services for the benefit of humans. It can also help to ensure long-term human well-being and sustainable development.

The difference in philosophy between anthropocentricity and biocentricity has led to a debate [15]. In terms of anthropocentric philosophy, the benefits that humans can derive from their activities are a major focal point. Anthropocentric philosophy tends to prioritize humanity and view other species only as exploitable resources. Exploitation or protection for human well-being is a primary choice and has attracted attention because of the self-interest of humans. Biocentricity seems an ideal, however. On behalf of sustainability, the evaluation conducted in the current research affirmed that conservation for hillside forests can ensure local residents’ welfare and so leads to a better Anthropocene.

### 4.3. Local Industries and Livelihood of Residents

Yilan is an agricultural county and attaches considerable importance to environmental protection. Limited industrial activity takes place in the coastal area. The land is fertile, and water and soil resources are pure and abundant.

Agricultural recreational activities are popular in this area, and they can be combined with forest ecological resources in the foothills to promote ecotourism and leisure agriculture in the form of organic agriculture or ecological or environmentally friendly agriculture. Recreational, organic, or friendly farming all depend on a healthy ecosystem. Yilan County is an ideal area to promote these farming practices and is an appropriate area for Taiwan to construct a model of sustainable agriculture.

### 4.4. Policy Implementation at the Mountain Frontier of Human Activities

The livelihood and property of local residents rely substantially on nearby hillside forests [57]. In this study, the local residents exhibited high agreement with the value of the ecosystem services, thus highlighting the importance of maintaining slope forest ecosystems for the welfare of local residents. Disturbing slope forests would strongly affect the economic well-being of local residents in relation to ecosystem services; thus, the current expansion of agriculture to hillside forest areas due to climate change is inappropriate. Local agriculture can be promoted through dedicated organic farming due to the availability of pristine water and soil and can be combined with the recreational or processing industry instead of advanced mechanized farming. Moreover, hillside ecotourism that can maintain the ecosystem services and sustain residents’ livelihood should be promoted.

### 4.5. Noncommodification of Nature: Economic Valuation from the Perspective of Ecological Economics

Market exchange systems can only represent a part of the economic value of ecosystem services. In human society, trading is a mechanism for maintaining social order. Most ecosystem services are public goods with positive externalities and are neglected by the public. The value of nonmarketable services cannot be directly presented in the trading market. Thus, the nonmarketable value of ecosystem services is often omitted in society, leading to lack of focus on such services in relevant policies. The value of nonmarketable services should be considered to understand the true economic value of the ecosystem from the perspective of the interdisciplinary field of ecological economics. 

The economic value of nonmarketable services should be estimated in order to increase the importance attached to the conservation of services offered by the natural environment. This would not be a valuation for subsequent sale in the human-centered misconception of the natural capital marketization and financialization. In the contingent valuation method, welfare is measured in terms of mainstream value reasoning in consumer theory. The value index can guide the conservation of the environment for public good and as a common asset; this valuation does not refer to a price for sale in the dynamic circulation of materials, energy, and services from the ecological economics perspective. In terms of geographical Anthropocene [68], humans are now overwhelming the great forces of nature. As aforementioned, conservation and protection of hillside forests in the mountain frontier is critical for a better Anthropocene.

## 5. Conclusions

At the frontier of land development and the junction of nature and human society, the forest ecosystem provides nearby residents with provisioning, regulating, culture, and supporting ecosystem services and generates economic value. Most of the ecosystem services are nonmarketable, and only a few of the services are marketized. No market price can accurately quantify the direct benefits that nearby residents derive from these services. In contemporary society, specific price indicators generated in the trading market system are used as tools for economic and social decision-making. Thus, ecosystem services have not been provided reasonable attention in the process of overall social decision-making. This study investigated how ecosystem services benefit residents in a frontier area from the perspective of ecological economics.

The single-bounded dichotomous contingent valuation method was applied with logit regression. The results indicated that the forests in the low-latitude hillsides of the Lanyang River Basin currently provide the local residents with the provisioning, regulating, culture, and supporting ecosystem services. The economic value of the four ecosystem services was considerably high. The results of this study highlight the importance of the conservation and maintenance of the low-latitude hillside forest ecosystem and that disturbances in the hillside forest can affect the economic well-being associated with the ecosystem services enjoyed by local residents. 

This study investigated the determinants of the value of these ecosystem services. Binary regression analysis was conducted to identify factors affecting the residents’ willingness to pay specific bidding amounts for ecosystem services. The results indicated that the residents’ willingness to pay increased with their agreement with the value of the ecosystem services and their income. The residents with experience in donating to environmental organizations tended to be willing to pay the amount asked. However, the older individuals with low income in the area had lower willingness to pay due to their perceptions of affordability.

Ecosystem services refer to benefits that ecosystems provide to humans. Taiwan is mountainous and densely populated. Thus, the appropriate management of forest cover at the intersection of natural ecosystems and areas with widespread human activity is crucial. The forest ecosystem service at the junction of agricultural and residential areas is extremely sensitive to the surrounding land use and social and economic activities as well as changes in nearby land-use patterns. Moreover, the societal disturbances can affect ecosystem services. The results of this study have crucial policy implications. The conservation of hillside forests and the performance of ecosystem services are directly related to the survival, life, and livelihood of local residents. The forests and surrounding residents dynamically and continually interact with ecosystem services. Individuals who live nearby hillside forests benefit from the ecosystem services provided by these areas; residents who engage in occupations dependent on nature stand to benefit the most from such services. The life of nearby residents is closely integrated with the provisioning, regulating, culture, and supporting ecosystem services of the hillside forest. Relevant land planning policies should consider the interaction between humanity and nature. Current hillside forests that are well protected by government zoning regulation are better maintained. Viable local policy for vicinity residents’ livelihood is suggested to promote ecotourism to maintain the ecosystem services of hillside forests and to integrate together with local recreational and organic agriculture.

## Figures and Tables

**Figure 1 ijerph-19-06193-f001:**
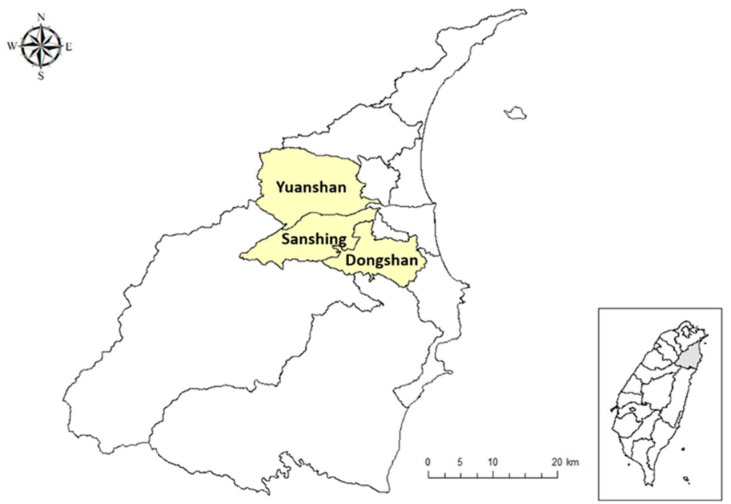
Study site: Sanshing Township (SST), Yuangshan Township (YST), and Dongshan Township (DST) in Yilan County, Taiwan.

**Table 1 ijerph-19-06193-t001:** Means and percentiles of the pilot survey.

Ecosystem Services	Provision	Regulation	Culture	Support
number of respondents	*n* = 50	*n* = 50	*n* = 50	*n* = 50
mean	2269	2389	1927	2460
percentile				
P20	211	223	500	223
P30	284	440	500	440
P50	1000	1000	1000	1000
P70	2000	2000	2000	2000
P80	3000	3000	3000	3000

Unit: New Taiwan Dollar.

**Table 2 ijerph-19-06193-t002:** Residents’ opinion regarding the ecosystem services provided by the hillside forest.

Likert Scale	5	4	3	2	1	Total
Ecosystem services						
provision	236 (53.15)	161 (36.26)	34 (7.66)	10 (2.25)	3 (0.68)	444 (100)
regulation	289 (65.09)	137 (30.86)	14 (3.15)	3 (0.68)	1 (0.23)	444 (100)
culture	247 (55.63)	164 (36.94)	25 (5.63)	8 (1.8)	0 (0.00)	444 (100)
support	254 (57.21)	161 (36.26)	26 (5.86)	2 (0.45)	1 (0.23)	444 (100)

Note: This table illustrates the number and percentage (in parentheses) for a total of 444 respondents. A 5-point Likert-type scale was used to measure the agreement of the residents with statements regarding the ecosystem services provided by the low-latitude forest in the vicinity. The responses for the 5-point scale were strongly agree, agree, neutral, disagree, and strongly disagree for the scores of 5, 4, 3, 2, and 1, respectively.

**Table 3 ijerph-19-06193-t003:** Residents’ opinions regarding the ecosystem services (by township).

Township	SST (*n* = 89)					YST (*n* = 136)					DST (*n* = 219)				
Likert Scale	5	4	3	2	1	5	4	3	2	1	5	4	3	2	1
Ecosystem services															
provision	55 (12.39)	29 (6.53)	5 (1.13)	0 (0.00)	0 (0.00)	79 (17.79)	50 (11.26)	5 (1.13)	1 (0.23)	1 (0.23)	102 (22.97)	82 (18.47)	24 (5.41)	9 (2.03)	2 (0.45)
regulation	60 (13.51)	25 (5.63)	3 (0.68)	1 (0.23)	0 (0.00)	96 (21.62)	36 (8.11)	3 (0.68)	0 (0.00)	1 (0.23)	133 (29.95)	76 (17.12)	8 (1.80)	2 (0.45)	0 (0.00)
culture	50 (11.26)	35 (7.88)	4 (0.90)	0 (0.00)	0 (0.00)	80 (18.02)	41 (9.23)	10 (2.25)	5 (1.13)	0 (0.00)	117 (26.35)	88 (19.82)	11 (2.48)	3 (0.68)	0 (0.00)
support	56 (12.61)	27 (6.08)	5 (1.13)	0 (0.00)	1 (0.23)	80 (18.02)	52 (11.71)	4 (0.90)	0 (0.00)	0 (0.00)	118 (26.58)	82 (18.47)	17 (3.83)	2 (0.45)	0 (0.00)

Note: This table illustrates the number and percentage for a total of 444 respondents.

**Table 4 ijerph-19-06193-t004:** Even distribution of the bidding amount in the three townships.

Townships	SST		YST		DST		Total	
Bid	Number	%	Number	%	Number	%	Number	%
250	18	20.22	28	20.59	42	19.18	88	19.82
500	17	19.10	27	19.85	42	19.18	86	19.37
1000	18	20.22	27	19.85	45	20.55	90	20.27
2000	18	20.22	27	19.85	45	20.55	90	20.27
3000	18	20.22	27	19.85	45	20.55	90	20.27
sum	89	100.00	136	100.00	219	100.00	444	100.00

**Table 5 ijerph-19-06193-t005:** Agreement to pay the corresponding bidding amount.

		Provision		Regulation		Culture		Support	
Bid	Number of Respondents	Number “Yes” Response	%	Number “Yes” Response	%	Number “Yes” Response	%	Number “Yes” Response	%
250	88	52	59.09	54	61.36	47	53.41	51	57.95
500	86	45	52.33	48	55.81	45	52.33	45	52.33
1000	90	47	52.22	47	52.22	44	48.89	46	51.11
2000	90	37	41.11	35	38.89	36	40.00	36	40.00
3000	90	32	35.56	32	35.56	29	32.22	32	35.56
Sum	444	213		216		201		210	
(%)		(47.97)		(48.65)		(45.27)		(47.30)	

**Table 6 ijerph-19-06193-t006:** Definitions of variables.

Variable	Definition	Mean (S.D.)			
		SST	YST	DST	Total
$P\left(Y\right)$	Binary dependent variable, used to represent the willingness to pay. In the survey, if the respondent indicated “yes”, *P*(*Y*) = 1; otherwise, it is 0.	-	-	-	-
C	Constant	-	-	-	-
BID	The bidding value (New Taiwan dollar, NTD)	-	-	-	-
${\mathrm{AGREE}}_{\mathrm{EE}}$	Agreement with the ecosystem services. *	-	-	-	-
GENDER	Gender dummy, 1, if female 0, if male	0.56 (0.50)	0.52 (0.50)	0.45 (0.50)	0.50 (0.50)
${\mathrm{OCCU}}_{D_{\mathrm{PUBLIC}}}$	Occupation dummy, 1, those who do not directly depend on local resources for their livelihood, such as those in the military, civil service, and education sector; 0, otherwise.	0.11 (0.32)	0.10 (0.31)	0.09 (0.29)	0.10 (0.30)
${\mathrm{STAY}}_{\mathrm{HR}}$	Average length of stay per visit (hours)	1.76 (1.12)	1.78 (1.16)	1.79 (1.15)	1.78 (1.14)
${\mathrm{AGE}}_{D60}\times {\mathrm{INC}}_{D30}$	${\mathrm{AGE}}_{D60}\times {\mathrm{INC}}_{D30}$ = 1 if older resident with low income; otherwise, the value is 0. ^#^	0.15 (0.36)	0.14 (0.35)	0.12 (0.32)	0.13 (0.34)
${\mathrm{INC}}_{10T}$	Yearly household income (10,000 NTD)	51.40 (33.95)	56.54 (37.56)	57.58 (32.82)	56.02 (34.56)
${\mathrm{DON}}_{D}$	Donation dummy 1, history of donation for environmental protection; 0, otherwise.	0.21 (0.41)	0.29 (0.45)	0.20 (0.40)	0.23 (0.42)

Note: Standard deviations are indicated in parentheses. * A typical 5-point Likert scale was used (5 = strongly agree, 4 = agree, 3 = neutral, 2 = disagree, and 1 = strongly disagree). ^#^ Two dummy variables were multiplied to create a variable (${\mathrm{AGE}}_{D60}\times {\mathrm{INC}}_{D30}$) representing older residents with low income. ${\mathrm{AGE}}_{D60}$ = 1, if age > 60 years; otherwise, the value is 0. ${\mathrm{INC}}_{D3}$ = 1 if the yearly household income is <300,000 NTD; otherwise, the value is 0.

**Table 7 ijerph-19-06193-t007:** Estimation results of the logit model.

Aspects of Ecosystem Services	Provision			Regulation			Culture			Support		
Variable	Coefficient		Prob.	Coefficient		Prob.	Coefficient		Prob.	Coefficient		Prob.
C	−2.3566	**	0.0013	−2.8351	**	0.0016	−2.4930	**	0.0017	−2.5103	**	0.0017
BID	−0.000372	***	0.0004	−0.000478	***	≦0.0001	−0.000388	***	0.0002	−0.000392	***	0.0002
${\mathrm{AGREE}}_{\mathrm{EE}}$	0.4985	***	0.0009	0.6071	**	0.0014	0.4488	**	0.0062	0.5151	**	0.0023
GENDER	−0.3321		0.1157	−0.4245	*	0.0460	−0.2621		0.2130	−0.3379		0.1060
${\mathrm{OCCU}}_{D_{\mathrm{PUBLIC}}}$	−1.1301	**	0.0017	−1.0476	**	0.0035	−1.0967	**	0.0028	−1.0541	**	0.0031
${\mathrm{STAY}}_{\mathrm{HR}}$	0.1631		0.0802	0.1975	*	0.0355	0.2161	*	0.0196	0.2006	*	0.0294
${\mathrm{AGE}}_{D60}\times {\mathrm{INC}}_{D30}$	−1.0980	**	0.0025	−1.0377	**	0.0034	−0.8493	*	0.0161	−1.0427	**	0.0030
${\mathrm{INC}}_{10T}$	0.0089	*	0.0109	0.0097	**	0.0065	0.0100	**	0.0040	0.0087	**	0.0098
${\mathrm{DON}}_{D}$	0.8491	**	0.0010	0.7813	**	0.0025	0.8886	***	0.0005	0.6833	**	0.0068

Note: *P*(*Y*) is the dependent variable. Refer to Table 6 for variable definitions. *, **, and *** significant at the 10%, 5%, and 1% levels, respectively. Standard errors are in parentheses.

**Table 8 ijerph-19-06193-t008:** Point estimates of the economic value of the ecosystem services. Unit: NTD.

Aspects of Ecosystem Services	Provision	Regulation	Culture	Support
	2688.17	2092.05	2577.32	2551.02

Note: Cameron [34,63] indicated that the point estimates of the willingness to pay in the single-bounded contingent valuation method can be approximated using the formula E(WTP) = $-1/\hat{\beta_{1}}$, and the $\hat{\beta_{1}}$ is the estimator of the bidding variable in logit regression.

## Data Availability

Not applicable.

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
