# Peer review of "Resident Willingness to Pay for Ecosystem Services in Hillside Forests"

_ijerph, 2022, doi:10.3390/ijerph19106193_

Round 1
Reviewer 1 Report
This article is seeming to be well-organized and structured and uses the appropriate methodologies, however in table 5 the words “re-sponded” have been moved which makes it less understandable. I did not notice any more errors in the text, pictures, or tables.
Author Response
Thanks for your suggestion and we are sorry for not spelling words well in Table 5. We have revised the Table 5 which is more understandable now. Thanks again for your comments. Please see our revised manuscript.

Reviewer 2 Report
This manuscript investigated an ecosystem’s economical value estimated by the neighboring residents and whether these residents are willing to actually pay for the ecosystem services. The four perspectives included the provisioning, regulating, cultural, and supporting ecosystem services, which were all evaluated considerably valuable by residents. Results also showed that the residents’ willingness to pay increased with their agreement with the value of the eco-596 system services and their income.
Comments:
The manuscript is written with relevant background information and particularly methodology about the topic. The discussion is well elaborated with many interesting perspectives. For data presentation, there are much valuable information that supports the conclusions. Yet if some tables could be presented as figures, information and message carried in this research would make a greater impact on the audience. In terms of English writing, the content was clearly and logically laid out. All research-related terms were explained. In summary, this research showed research methods and perspectives that could be adapted for other similar sites.
Author Response
Thanks for your suggestion and comments. We choose applying the tables to show our research results due to the numerous information and data which are not easy to present perfectly in figures. Hope you could agree us continually using tables. Thanks again for your comments.

Reviewer 3 Report
The topic of the manuscript is of great interest for the social sciences and I think that could be useful both for the scientific communities and local governments. The sentences in the first lines (28-31) are quite simple and disconnected, that it is a pity considering that the rest of the manuscript is well written with a very good syntaxes. I suggest to write a more complex sentence in line with the rest of the text.
In addition I suggest to the author to prepare some figures (2 or 3) presenting the data reported in the tables 2-3-4-5 (e.g box plot and/or pie charts). Figures are more self-explanatory than tables and they will give impact to the manuscript. The authors can decide which tables are the most useful to represent as figure. To increase the impact of the manuscript and gain more visibility I also recommend a graphical abstract. However, I know that it may be very difficult to prepare so this is only a simply suggestion for the authors. But I strongly recommend the rest of the figures.
Author Response
Thanks for your suggestion and comments.
- We have revised the sentences in the first lines (28-31) in Introduction as following. “It is possible to conceive that our world is highly developed, however, the planetary ecosystem and resources are finite. Because the economic system is a subsystem in the planetary boundary [6,7], the finite nature of Earth's resources limits the economic growth.”
- We choose applying the tables to show our research results due to the numerous information and data which are not easy to present perfectly in figures. Hope you could agree us continually using tables. Thanks again for your comments.
